# Laser Treatments for Improving Electrical Conductivity and Piezoresistive Behavior of Polymer–Carbon Nanofiller Composites

**DOI:** 10.3390/mi10010063

**Published:** 2019-01-18

**Authors:** Andrea Caradonna, Claudio Badini, Elisa Padovano, Antonino Veca, Enea De Meo, Mario Pietroluongo

**Affiliations:** 1Department of Applied Science and Technology, Politecnico di Torino, Corso Duca degli Abruzzi 24, 10129 Torino, Italy; andrea.caradonna@polito.it (A.C.); elisa.padovano@polito.it (E.P.); mario.pietroluongo@polito.it (M.P.); 2CRF, Centro Ricerche FIAT, Strada Torino 50, Orbassano, 10043 Torino, Italy; antonino.veca@crf.it (A.V.); enea.de-meo@external.fcagroup.com (E.D.M)

**Keywords:** carbon nanofillers, electrical conductivity, piezoresistive behavior

## Abstract

The effect of carbon nanotubes, graphene-like platelets, and another carbonaceous fillers of natural origin on the electrical conductivity of polymeric materials was studied. With the aim of keeping the filler content and the material cost as low as possible, the effect of laser surface treatments on the conductivity of polymer composites with filler load below the percolation threshold was also investigated. These treatments allowed processing in situ conductive tracks on the surface of insulating polymer-based materials. The importance of the kinds of fillers and matrices, and of the laser process parameters was studied. Carbon nanotubes were also used to obtain piezoresistive composites. The electrical response of these materials to a mechanical load was investigated in view of their exploitation for the production of pressure sensors and switches based on the piezoresistive effect. It was found that the piezoresistive behavior of composites with very low filler concentration can be improved with proper laser treatments.

## 1. Introduction

### 1.1. Polymer/Carbon Filler Conductive Composites 

Polymer composites with carbonaceous micro and nanofillers were widely investigated during recent years because of their mechanical, electrical, and thermal properties. In particular, nanocomposites show noticeable structural and functional properties that can be exploited for a broad range of applications in every field. The potential of these materials as low-weight structural materials and functional materials for optical devices, electromagnetic shields, electric components, and medical devices (body-attachable adhesives for measuring biosignals) attracted increasing interest [1,2,3,4,5].

Actually, carbon nanofillers can be used for greatly enhancing the electrical conductivity of both thermoplastic and thermoset resins and improving their mechanical behavior at the same time. For these reasons, the potential of carbon nanotubes (CNTs) and graphene-like nanoplatelets (GNPs) for processing conductive polymer-based composites was investigated [1,2,3,4]. The electrical conductivity of CNTs ranges between 10^5^ and 10^7^ S/m, and the typical conductivity value for GNPs is around 10^5^ S/m. These conductivity values are similar to those of graphite (10^5^ S/m on the graphene sheets constituting the graphite structure), which is traditionally used as filler for polymeric composites. On the other hand, the high aspect ratio of CNTs and GNPs can be exploited for achieving more easily the percolation threshold for electrical conductivity. This threshold for electrical conductivity is observed when the concentration of carbon filler is high enough to allow the formation, inside the polymer matrix, of a continuous network of filler particles, which causes a sudden jump in electrical conductivity. 

However, the electrical conductivity of polymer–nanocarbon composites is also greatly affected by the kind of matrix and the fabrication process. For instance, a comparison of literature data [1,5] shows that the percolation path forms after the addition of very different amounts of CNTs to different matrices (from less than 1 wt.% to more than 10 wt.%). Likewise, the addition of similar loads of GNPs to different matrices, as well as the fabrication of the same composite with different methods, can result in dramatically different electrical conductivity values, placed in a range even ten orders of magnitude wide [3]. As a consequence, the filler concentration required for achieving the threshold depends not only on the kind of composite, but also on the composite processing method. In fact, very different values of filler concentration at the percolation threshold were reported in the literature for composites with the same matrix and filler [1,2,3], which makes it very hard to establish a filler concentration of general validity corresponding to the threshold. 

### 1.2. Relevance of Filler Morphology, Synergetic Effects, and Laser Treatments on Electrical Conductivity

Huang et al. [6] modeled the effect of the geometric factor on the conductivity of composites containing CNTs and GNPs. In general, due to the higher aspect ratio, CNTs are believed more effective than GNPs for achieving the percolation threshold [7,8], but the method adopted for the fabrication of the composite in this case also entails great importance. In fact, the alignment of CNTs along a direction favors the formation of a conductive pathway for both electrons and phonons and, therefore, can be exploited for enhancing conductivity. Goh et al. [9] reviewed the methods that can be used for obtaining a preferential orientation of fillers (e.g., the application of electrical or magnetic fields and shear forces during the composite fabrication). Both the orientation and size of GNPs were also found to appreciably affect the electrical, thermal, and mechanical behavior of polypropylene–GNP composites [10]. The combination of fillers with different aspect ratios can be exploited to obtain a conductive network inside a polymeric matrix. A synergetic effect between CNTs and GNPs or CNTs and carbon black particles was observed [11,12] for epoxy and some thermoplastic matrices, such as styrene–butadiene, poly(ethersulfone), polyvinylidene fluoride, and poly(vinyl alcohol). However, it is doubtful that the synergetic effect can occur in every kind of composite, since Paszkiewicz et al. [13] did not find any evidence of it when investigating the electrical conductivity of polyethylene filled by CNTs or CNTs plus GNPs. 

According to an alternative approach, surface laser treatments proved their suitability for locally improving the electrical conductivity of nanocarbon-filled polymers and, thus, for processing metal-free electrical circuits [14,15,16]. The laser beam causes the pyrolysis of the polymeric matrix on the composite surface with the formation of gaseous species that leave the material. In this manner, the filler/matrix ratio greatly increases, the percolation threshold is locally achieved, and conductive tracks form on the surface of an insulating composite with low mean filler content. 

### 1.3. Polymer/Carbon Filler Piezoresistive Composites 

The addition of carbon fillers shows potential not only for converting insulating polymers into conductive materials, but also for providing polymers of antistatic, electromagnetic absorption, and piezoresistive properties. 

The piezoresistive behavior of polymers filled by CNTs, GNPs, thermally reduced graphene oxide, or carbon nano-blocks can be exploited for the fabrication of strain sensors [5,17,18,19,20]. The piezoresistive effect in these nanocomposites can be due to changes in the network formed by filler and variation of the piezoresistivity of fillers because of their own deformation. The stability and reproducibility of the electro-mechanical response of these nanocomposites is required for practical applications and should be further investigated. Systems with rather high filler content (from 5 to around 20 wt.% of filler) were studied [17,20], while efforts should be spent to keep the filler concentration and the material cost as low as possible.

### 1.4. Aim of the Work

In this paper, the laser processing of conductive tracks on several nanocomposites produced in a very similar manner was investigated, with the aim of demonstrating the viability of this approach for most composite systems, investigating the importance of laser writing parameters, and reducing the filler concentration needed for achieving locally electrical conductivity. The piezoresistive behavior of some nanocomposites and the effectiveness of laser treatment for improving the piezoresistive properties were also studied.

## 2. Materials and Methods 

Several commercial thermoplastic polymers were used as composite matrices: high-density polyethylene (HDPE; Lupolen 4261 A IM, LyondellBasell, Houston, TX, USA), polypropylene–ethylene copolymer (PP; Hostacom CR 1171 G1, LyondellBasell), polycarbonate and acrylonitrile–butadiene–styrene blend (PC-ABS; Babyblend T65XF, Bayer Material Science-Covestro, Leverkusen, Germany), acrylonitrile–butadiene–styrene (ABS; Cycolac, Sabic, Riyadh, Saudi Arabia), and ethylene–propylene–diene monomer (EPDM; 719.A65 Forflex, SO.F.TER, Lebanon, TN, USA). These matrices were blended with carbonaceous micro and nanofillers: multiwall carbon nanotubes (MWCNTs; NC7000, Nanocyl, Sambreville, Belgium), graphene nanoplatelets (GNPs 25; AB304024 25 µm wide and 6–8 nm thick, ABCR Gute Chemie; GNPs grade 4, 1–2 µm wide and less than 4 nm thick, Cheaptube Inc., Cambridgeport, VT, USA), graphite flakes with a median size of 7 to 10 µm (Alfa Aesar, Ward Hill, MA, USA), and biochar OSR700 (UK Biochar Research Center, Edinburgh, Scotland). Also, masterbatches of composite materials were used to produce the samples under investigation. A masterbatch PP/15 wt.% CNTs (Nanocyl NC7000) was acquired from Hostacom, while other masterbatches (HDPE/6 wt.% CNTs, HDPE/12 wt.% GNPs 25, PC-ABS/2.75 wt.% CNTs, and HDPE/12 wt.% graphite) were produced in the laboratory by mixing the matrix and the filler. These masterbatches were processed by melt blending using a PlastiCorder Brabender W50E model. The material out of the mixer was then transferred to a pelletizing machine (Piovan RSP 15/15, Maria di Sala, Italy). Several composite samples with different compositions were finally prepared according to two possible paths: direct mixing of matrix and filler, or mixing of one or two masterbatches with the unfilled matrix. These composite samples were processed by mixing pellets of the starting materials using a twin-screw extruder (EuroLab 16mm XL 40:1 L/D Thermo Fisher Scientific, Waltham, MA, USA) with a final pelletizing unit (Thermo Haache Eurolab). The processing path for each material submitted to the investigation of electrical behavior is summarized in Table 1. Composite plates (67 mm × 12.8 mm × 2.9 mm) were fabricated by injection molding (Babyplast equipment of Cronoplast S.L.) of the composite pellets produced as described above. Larger plates (140 mm × 90 mm × 3 mm) were also produced (using a press Micro 65 series, Sandretto, Pont Canavese, Italy) with the aim of investigating the possible interaction between adjacent tracks.

Laser tracks were processed on the surface of these plates by using a CO_2_ laser equipment (LASIT Towermark XL (Torre Annunziata, Italy), with a power of 100 W, a wavelength of 10.6 μm, and a spot size of 100 µm with 0 defocusing). Several parallel tracks were written at a distance of 1 cm on the plates in order to investigate their electrical resistance, as well as the inter-track resistance. A single track was produced in the middle of the plates in order to investigate the piezoelectric behavior. The laser treatment was performed under nitrogen atmosphere in order to avoid sample oxidation. The parameters adopted for the laser treatment were optimized for the different composites. 

The morphology of the tracks was investigated using a field-emission (FE)-SEM Zeiss MERLIN (Carl Zeiss AG, Oberkochen, Germany) and a Profilometer confocal microscope Leica DCM8 (Leica Microsystems Inc., Buffalo Grove, IL, USA).

Electrical resistance of the as-produced composites and tracks processed by laser writing was measured using a multimeter (Keitley 2700E, full scale value 120 MΩ, Keithley Instruments, Cleveland, OH, USA). Silver paint was deposited at the beginning and the end of the tracks with the aim of granting better contact with the steel probes of the multimeter. 

For the investigation of the piezoelectric behavior, cyclic three-point bending tests were carried out using a dynamometer (Instron 5544, Norwood, MA, USA), contemporaneously measuring the displacement and the resistance variation by means of an extensometer and a Keithley 2700E multimeter. The software of the dynamometer (Bluehill3, Instron, Norwood, MA, USA) and that of the multimeter (Labview, version 2015, National Instruments, Austin, TX, USA) were interfaced with the data recording system.

## 3. Results

### 3.1. Laser Writing of Conductive Tracks on Carbon-Filled Polymers 

Laser treatment of the surface of insulating composites can be exploited for locally changing their composition with the aim of increasing the carbon filler concentration until the percolation threshold for electrical conductivity is achieved. This approach offers the opportunity of creating conductive paths through the modification of the surface of composites that are not conductive and are not too expensive, because they contain rather low concentrations of expensive fillers, like CNTs and GNPs, or higher concentrations of cheap fillers.

In every case, the laser beam causes the pyrolysis of the polymeric matrix, which results in the formation of gaseous species that leave the material, thereby increasing the filler/matrix ratio. Laser irradiation was also exploited for improving the nanostructure of films made of CNTs deposited on different substrates [21,22,23]. This treatment resulted in the change of the film resistance owing to the decrease in impurity content in CNTs and defect healing/recrystallization. Therefore, in principle, laser irradiation could provide some conductivity improvement through filler modification. On the other hand, the treatment conditions we adopted for writing the conductive tracks seem very far from those suitable for modifying the structure of carbonaceous nanofillers (for instance, in this last case, the laser wavelength was one order of magnitude lower than that we used in the present investigation). For this reason, the change in filler concentration resulting from the polymeric matrix depletion can be considered as the main effect causing the enhancement of conductivity inside the tracks produced by laser writing.

The effect of the laser treatment is strictly related to the amount of energy given to the substrate, which in turn depends on the kind of laser and the processing parameters. Laser power and frequency, speed of movement of the laser beam on the surface, distance of the laser source from the treated surface (called defocusing hereafter), and numbers of laser runs on the same part of the surface (number of treatment repetitions) were the main parameters to be considered. The final electrical properties of the tracks that the laser beam writes on the surface also depend on the characteristics and the concentration of conductive filler, as well as on the tendency of the polymeric matrix to undergo thermal decomposition.

The concept of laser surface treatment and the morphology of the surface conductive track are shown in Figure 1. The laser writing on the composite surface locally causes a great enhancement of concentration of the carbonaceous filler or fillers, represented in Figure 1 as green rods and blue particles dispersed within the matrix. This effect is confined within the track and affects a surface layer of the material 1–2 mm wide and some hundreds of micrometers thick. The surface morphology after the laser action is also depicted at different magnifications in Figure 1, as shown in the middle of the track where the laser leaves a forest of nanotubes protruding from the polymeric substrate. For practical applications, the conductive tracks must be processed inside a non-conductive support in order to avoid short circuits. 

In order to better understand the importance of the different processing parameters and their effect on the track conductivity, it is necessary to perform several tests, preferably using the design of experiment (DOE) approach, since it allows limiting the number of experiments [24]. As an example, the outcomes of the DOE approach in the case of laser treatment of polycarbonate/acrylonitrile butadiene styrene (PC/ABS) composite with 0.75 wt.% of CNTs are reported here. As the number of factors affecting the track conductivity was above four, a two-level fractional factorial design approach was adopted. Each laser trial differed from the others because two parameters out of five were changed. The lower and upper limit for the processing parameters were as follows: 5–30% of the maximum laser power, 100–600 mm/s for writing speed, 5–30 Hz for frequency, 0–50 for the number of repetition, and 0–50 mm for defocusing. The importance for conductivity enhancement of the different parameters and of their combinations is summarized by the Pareto plot in Figure 2. On the *y*-axis, each laser parameter or combinations of parameters possibly affecting the final conductivity of the tracks are reported. For each of these parameters or couple of parameters, the higher or lower importance of the effect on the final conductivity is represented by an index reported on the *x*-axis. 

From this plot, it is clear that the resistance chiefly depends on power and writing speed, as well as from the combination of these to parameters, because they determine the amount of energy which is given to the substrate in the unit of time. The number of repetitions affects the final resistance less, while the other parameters exert only secondary effects. Nonetheless, many repetitions of the treatment along the same track result in the progressive depletion of the matrix and, as a consequence, in the increase of filler concentration and conductivity. The effect of the number of repetitions on the resistance of tracks processed on the surface of PP–matrix composites with different loads of CNTs (from 1 wt.% to 4 wt.%) is depicted in Figure 3. For each curve reported in this figure, the measure of resistance was repeated on the same track after progressively increasing the number of laser runs. The resistance decreased more or less quickly with the number of repetitions depending on the starting conductivity of the composite, which is initially controlled by the filler concentration.

The main importance of these three parameters (power, writing speed, and number of repetitions) was observed for many kinds of polymer–carbon composites submitted to laser functionalization, but there are also limits in the selection of both material and processing conditions that should be considered for practical applications. As a matter of fact, the conductive tracks can be obtained by laser-treating composites with filler concentrations below or above the percolation threshold. When the filler concentration is over the percolation threshold, the conductivity inside the tracks is very good, but to create conductive paths inside a conductive material has no practical relevance because, of course, short circuits form between the tracks. The amount of energy delivered by laser treatment by unit of surface and unit of time, which increases when power increases and writing speed decreases, greatly affects the conductivity and the morphology of the tracks. However, only a fraction of the maximum power of the laser can be used, because too much energy results in deep tracks, high temperature gradients, and high thermal stresses, which can even cause deformation of the sample. Similar effects can be observed when the writing speed is excessively low. Then, these two parameters can be changed only within limited ranges and not independently. Several repetitions of the laser passage can be adopted to deliver the energy over a longer period, thus reducing the risk of sample distortion. However, the repetition of the treatment has a noticeable effect on the morphology of the track, since it causes an increase in depth and width of the conductive path. The variation of track morphology with the number of repetitions can be assessed using profilometry, as shown in Figure 4a,b.

Furthermore, defocusing is responsible for the track morphology. In Figure 4c, the profile of two tracks obtained on PP/2 wt.% CNT composites, with defocusing of 50 mm and 150 mm, are compared. The increase in defocusing (which must be coupled with the power increase, since the energy is spread on a larger surface) results in the track widening.

The widening of the tracks obtained on insulating composites can cause short circuits between adjacent tracks. This effect can be attributed to the presence of zones with increased filler concentration placed in between two tracks, resulting from an incomplete homogeneous dispersion of filler in the matrix. 

Conclusively, the best processing parameters should be selected not only with the aim of improving the conductivity as much as possible and obtaining reproducible resistance values, but also taking into account that inter-track conductivity and material deformation must be avoided. Nevertheless, the choice of the most suitable processing parameters mainly depends on the characteristics of the filler (such as intrinsic electrical conductivity, size, and aspect ratio), the filler load, the homogeneity of filler distribution inside the matrix, and the response of the matrix to the laser action. For these reasons, the laser process should be tailored for each kind of composite. Processing parameters selected for maximizing conductivity are shown for several different matrix/filler systems in Table 1. Concentrations of fillers below the percolation threshold were generally adopted for the composite production, and the laser writing process was tailored to increase the track conductivity as much as possible without causing short circuits between the tracks. On the other hand, the percolation thresholds depend on the kind filler and matrix, and the effectiveness of the production process of the composite; therefore, very different CNT loads (from 0.5 wt.% to 4 wt.%) were used for different matrices. In addition, for composites with a filler load around (PP/2 wt.% CNTs, HDPE/4 wt.% CNTs) or even over (HDPE/6 wt.% CNTs), the percolation threshold [1] was treated in order to better understand the impact of the filler concentration on the conductivity of the paths produced by the laser action. 

Laser power between 5% and 20% of the maximum power available and writing speed between 50 mm/s and 300 mm/s were adopted. Different combinations of these two parameters were used for different kinds of composites, which means that different amount of energy were required for the pyrolysis of different matrices, and that the concentration and kind of fillers can affect the energy adsorbed by the composite. The number of repetitions and the defocusing were tuned in order to obtain tracks of similar morphology on composites with different composition. Table 1 also shows that, in spite of the parameter optimization, the laser-writing process gives rise to conductive tracks showing very different resistance when different composite systems are treated. It is very hard to measure exactly the cross-section of each track owing to its irregular shape at a microscopic level. In addition, some conductive behavior of the material close to the track cannot be excluded. In fact, inside the thermally affected areas, some microstructure modification should also occur. For this reason, resistivity value was not calculated, but the measured resistance (which is not constant in every part of the track profile) was normalized with reference to the length unit of the track. Nonetheless, Table 1 shows that several polymeric matrices filled with CNTs or GNPs can be successfully submitted to laser ablation for the production of conductive tracks, and suggests some conclusions. When the CNT content inside each kind of matrix increased, the conductivity also generally increased. However, the opposite trend could be observed in some cases (see PC-ABS composites), very likely because CNTs can be more hardly dispersed in some matrices. When CNTs agglomerate and form bundles inside the matrix, the laser parameters must be changed in order to avoid inter-track conduction, and this can result in a worsening of the track conductivity. In addition, the variation of the filler load can require an adjustment of the processing parameters because the material response to the laser action depends on the filler/matrix ratio. The adoption of the same content of CNTs can result in different resistance of the tracks processed in the best manner on composites with different matrices. For instance, the track resistance was 12.3 kΩ/cm and 3.96 kΩ/cm for PP/1 wt.% CNTs and PC-ABS/1 wt.% CNTs, respectively. GNPs seemed less effective than CNTs when used alone or when they replaced part of the CNTs in the composite. The combination of a second filler with CNTs seemed more convenient when using graphite instead (see HDPE/4 wt.% MWCNTs/4 wt.% graphite system in comparison with HDPE/4 wt.% MWCNTs/4 wt.% GNP composite). A very high concentration of low-cost carbon particles was necessary to observe some conductivity in the tracks. This was the case of biochar, whose composite with 30 wt.% in a PP matrix showed laser tracks with poor conductivity. On the other hand, the resistance of these tracks was about six orders of magnitude lower than that of the unfilled matrix, which suggests that the laser treatment can also be exploited in this case to obtain antistatic properties. 

### 3.2. Piezoresistive Behavior

The piezoresistive behavior of composites with a EPDM/PP matrix (60/40 weight ratio) and CNTs was investigated. Concentrations of CNTs from 1 wt.% to 5 wt.% were used. The percolation threshold was found around 3 wt.% CNTs (Figure 5). This percolation threshold found for the blend between the ethylene–propylene–diene monomer and polypropylene seems consistent with the literature [1,12,25], which reports that concentrations of CNTs of 2 wt.%, 7.5 wt.%, or 7 vol.% are necessary for achieving percolation when PP, PE, or commercial thermoplastic elastomer based on EPDM/PP, respectively, are used as composite matrices. 

The piezoelectric behavior of such composites could be exploited for processing pressure sensors or switches based on the change in electrical resistivity occurring after deformation. The resistance of all composites with different filler loads progressively increased with the displacement occurring in the elastic and plastic fields when a flexural force was applied. However, only reversible deformations can be considered for practical applications, and then the displacement should be limited to the elastic field. The maximum displacement occurring in the elastic field was measured by a bending test. The stress/displacement curves obtained from three-point bending tests showed that the maximum elastic displacement of these composites increased with the increase in CNT concentration. On the other hand, only a little variation in resistance resulted from bending the material up to the elastic limit for samples containing 1 wt.% or 2 wt.% CNTs.

On the contrary, samples containing from 3 wt.% to 5 wt.% CNTs showed appreciable and almost linear resistance variation with displacement, but a good reproducibility of the piezoelectric effect was observed only for displacement over 1 mm. Figure 6 shows, for the composite with 4 wt.% CNTs, the typical variation of resistance with the cyclic change of displacement from 0 to 1.5 mm with a speed of 50 mm/min. It is possible to observe noise in the resistance signal when the deformation is recovered during each cycle, probably due to a rearrangement of CNTs inside the matrix occurring at the microscopic level.

However, the noise was greatly reduced when the maximum displacement increased to 2.5 mm (Figure 7).

The reproducibility of piezoresistive response to stress during long periods of cycling was also investigated; the resistance of unloaded material, as well as the maximum resistance change resulting from load application and consequent displacement, changed after a few hundreds of cycles; however, afterward, they remained constant with an increase in the number of cycles (Figure 8, Table 2). These outcomes prove that the composites under investigation can be exploited as pressure sensors, granting response to mechanical load for long periods.

As laser surface treatment is able to greatly improve the conductibility of carbon-based composites, the piezoresistive behavior was also tested on specimens with conductive tracks processed on the surface. Samples with a very low concentration of CNTs in a PC-ABS matrix were used for this investigation. Conductive tracks were processed in the middle of composite bars which were then submitted to cyclic flexural deformation, while their deformation and the resistance of the laser track were measured. Samples with CNT loads of 0.5 wt.%, 0.75 wt.%, and 1.0 wt.% were tested. The parameters for laser treatment were selected with the purpose of not decreasing too much the electrical resistance. For instance, the following parameters were adopted for the functionalization of the PC-ABS/0.5 wt.% CNT sample: P = 5%, scan rate = 600 mm/s, F = 5 kHz, and defocus = 50 mm. The electrical resistance of the track was 464 kΩ/cm, 172 kΩ/cm, and 76 kΩ/cm for composites with 0.5 wt.%, 0.75 wt.%, and 1.0 wt.% CNTs, respectively. The specimen resistance variation was measured during 2000 cycles of deformation up to a maximum displacement of 0.5 mm. Typical change of resistance during cycling is depicted in Figure 9 and Figure 10.

A resistance variation of 0.5% occurred during each cycle for the three kinds of composites, irrespective of the nominal concentration of filler. A precise correlation between displacement and resistance was always observed. Furthermore, the electrical signal was more sharp and regular when the filler concentration was 0.5 or 0.75 wt.%, while the composite with 1 wt.% CNTs gave more irregular resistance curves. Therefore, in the case of laser-functionalized piezoresistive composites, there is no reason to use samples showing enhanced filler content and conductivity, contrary to that which happened for non-laser-treated composites. In fact, piezoresistive behavior can also be achieved in the case of composites with low filler content by exploiting laser functionalization. 

## 4. Conclusions

Laser treatment was successfully used to produce conductive tracks on the surface of several kinds of polymer–nanofiller composites with a filler content lower than the percolation threshold for electrical conduction. The effectiveness of this functionalization process for writing conductive tracks inside composites with low loads of conductive filler was proven. In general, the final resistance of these tracks depends on the kind of filler and matrix, and the laser processing parameters. Conductive tracks can be more easily obtained when using CNTs, while, to obtain the same result, a higher concentration of GNPs is required. Carbonaceous micro-fillers like graphite or biochar can also be used as fillers. The addition of a small quantity of graphite to the composite allows reducing the content of the CNTs because of a synergetic effect between the two fillers, thus making the material cheaper. When using biochar, even in rather high concentrations, only antistatic properties can be achieved with the laser treatment. Very different resistance values were observed when laser-treating the surface of polymer composites showing different matrices filled with the same or similar concentrations of CNTs. 

Generally, the conductivity of laser tracks increased with the load of conductive filler, but there is not always a reason to increase the filler content, since its increase can cause agglomeration of filler particles, which results in a non-homogeneous filler distribution and, thus, a worse result of the laser treatment. 

The processing parameters of laser treatment should be optimized for each kind of composite, depending on the composition, and the main parameters to be optimized are power, writing speed, and number of repetitions. The parameter optimization is limited by side effects such as distortion of the sample and excessive enlargement of the tracks, which causes short circuits. Moreover, other parameters that have little influence on the conductivity show a non-negligible effect on these side effects. 

The addition of CNTs to a thermoplastic polymer also gives rise to piezoresistive behavior, which could be exploited for the fabrication of pressure sensors. The resistance variation with mechanical load and displacement depends on the CNT concentration. A good response to load was observed for EPDM/PP/CNT composites, but only for filler concentration exceeding 2 wt.%. Noise of the electrical signal was observed when the displacement was too low, but displacements over 1.5 mm were sufficient to overcome this drawback. The composite material with piezoresistive behavior can show some initial instability in terms of the range of resistance variation and average resistance value, probably due to a rearrangement of microstructure during the initial cycles of loading. However, the response of the materials soon stabilizes.

A surface laser treatment improved the piezoresistive behavior of PC-ABS/CNT composites. These composites with very small CNT concentrations (below the percolation threshold) showed good and stable piezoresistive behavior after the laser treatment. This characteristic offers potential for processing in situ sensors and switches using laser treatment of non-conductive composites with very low filler load.

## Figures and Tables

**Figure 1 micromachines-10-00063-f001:**
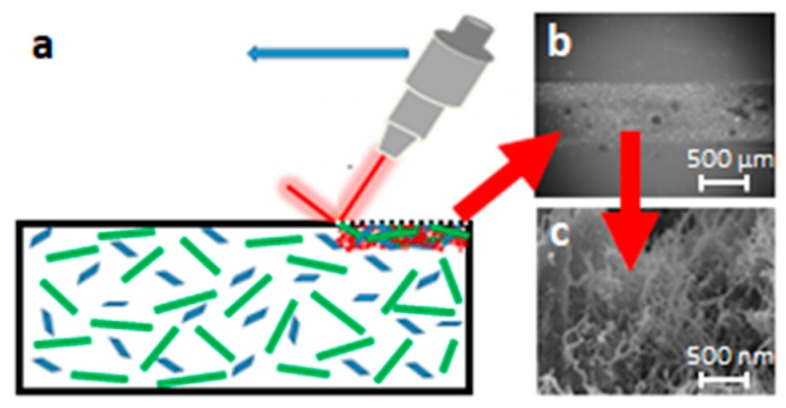
Effect of laser ablation on the surface (**a**); SEM micrographs at different magnifications of resulting tracks showing increased content of conductive filler (**b**,**c**).

**Figure 2 micromachines-10-00063-f002:**
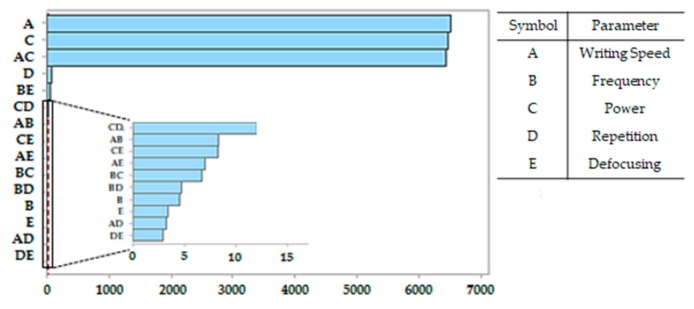
Pareto plot showing the relevance of different parameters and their combinations on the conductivity of tracks.

**Figure 3 micromachines-10-00063-f003:**
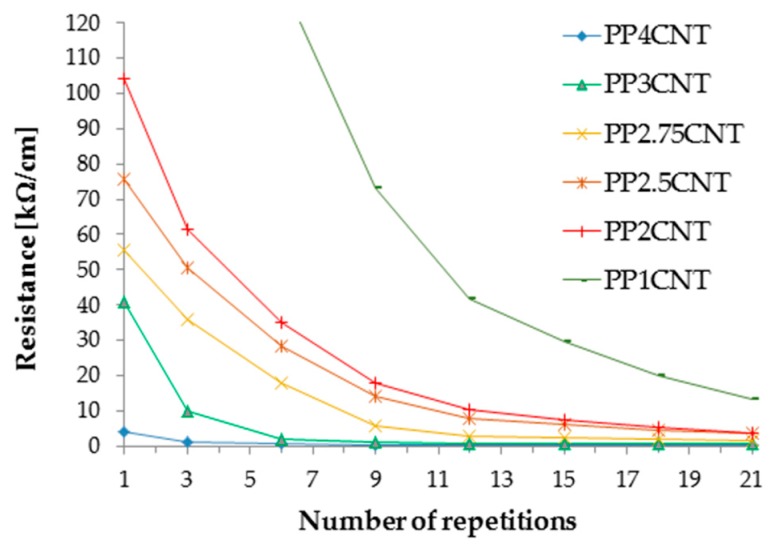
Decrease in track resistance with the number of laser treatments performed on polypropylene–ethylene copolymer (PP)/carbon nanotube (CNT) composites (P = 35%, scan rate = 200 mm/s, F = 15 kHz, defocus = 0 mm).

**Figure 4 micromachines-10-00063-f004:**
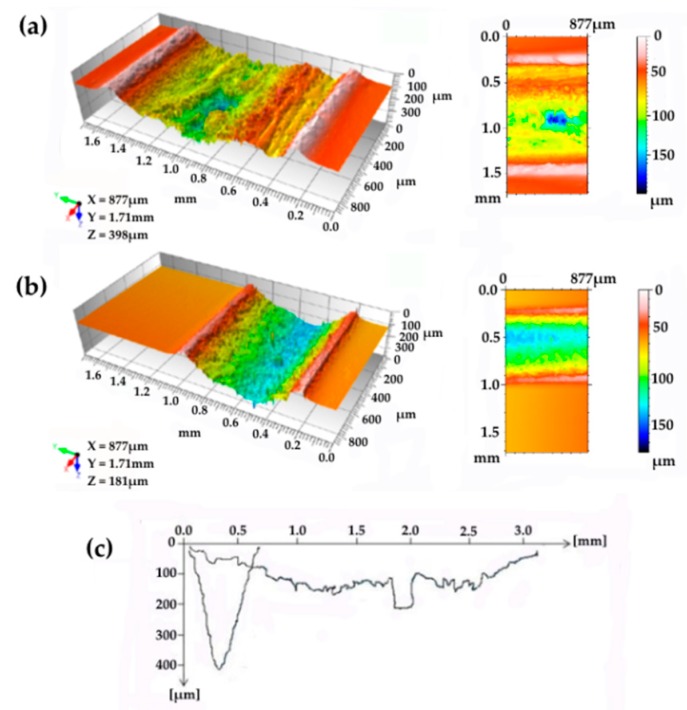
Profilometric measurements of tracks processed on polycarbonate and acrylonitrile–butadiene–styrene blend (PC-ABS)/0.5 wt.% CNT composite under different conditions: (**a**) wide and deep track resulting from repetitions (D = 0, P = 10, F = 5, S = 100, N = 20), (**b**) less severe treatment (D = 50, P = 50, F = 5, S = 600, N = 1), (**c**) comparison between narrow and deep track profile, obtained on PP/2 wt.% CNTs, resulting from low defocusing (D = 50, P = 15, F = 15, S = 200, N = 25) and wide track profile (D = 150, P = 40, F = 15, S = 200, N = 25).

**Figure 5 micromachines-10-00063-f005:**
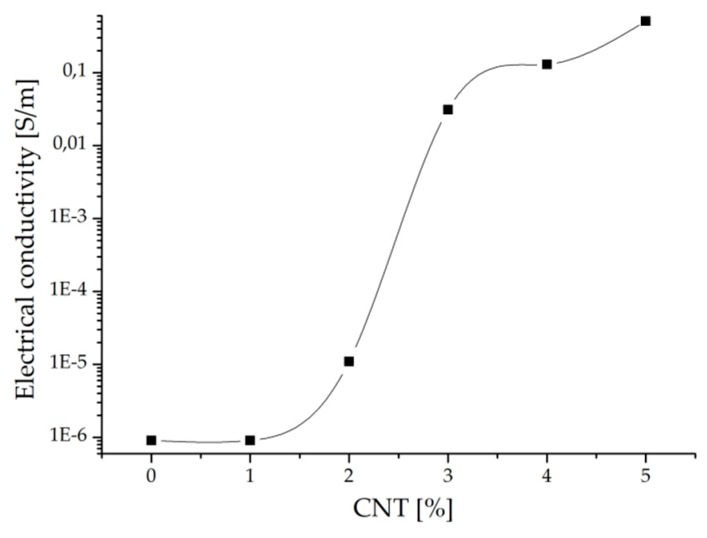
Percolation curve for ethylene–propylene–diene monomer (EPDM)/PP/CNT composites.

**Figure 6 micromachines-10-00063-f006:**
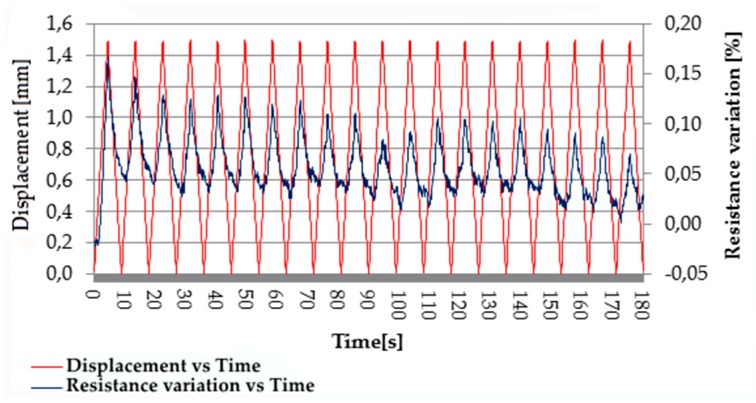
Piezoresistive behavior of EPDM/PP/4 wt.% CNT composite: cyclic resistance variation due to cyclic deformation from 0 to 1.5 mm.

**Figure 7 micromachines-10-00063-f007:**
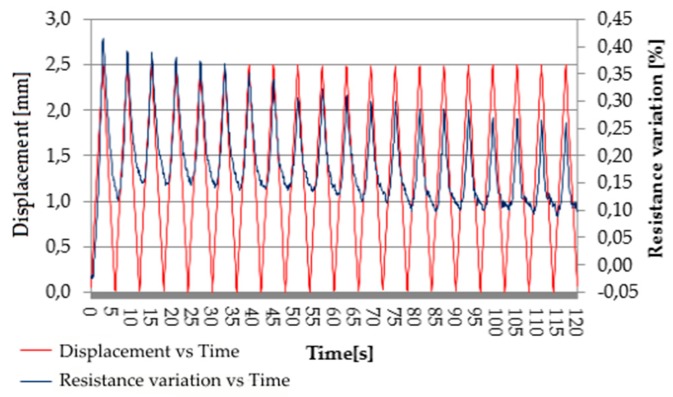
Piezoresistive behavior of EPDM/PP/4 wt.% CNT composite: cyclic resistance variation due to cyclic deformation from 0 to 2.5 mm (displacement speed: 50 mm/min).

**Figure 8 micromachines-10-00063-f008:**
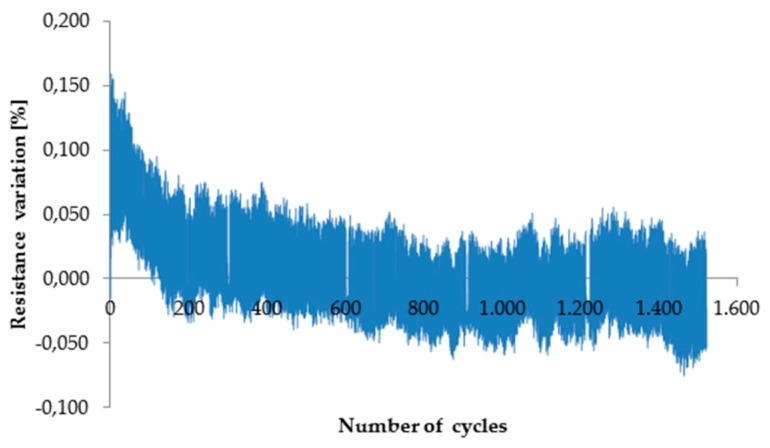
Testing of EPDM/PP/4 wt.% CNT composite up to 1500 cycles (displacement: 0–1.5 mm, speed: 50 mm/min).

**Figure 9 micromachines-10-00063-f009:**
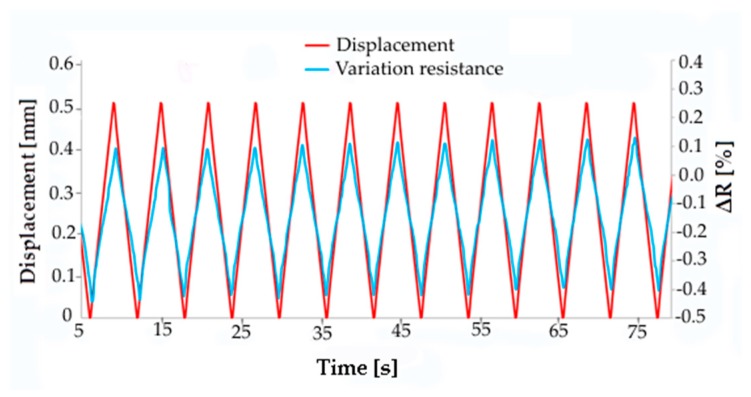
Displacement and resistance variation recorded during cycling of PC-ABS composite with 0.5 wt.% CNTs.

**Figure 10 micromachines-10-00063-f010:**
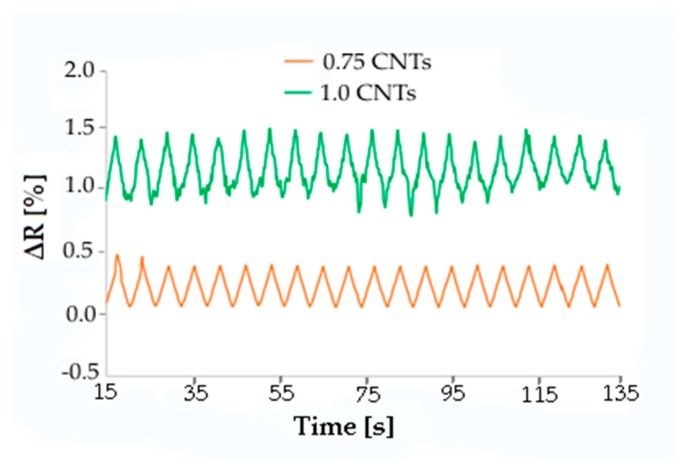
Resistance change during flexural fatigue cycles with maximum displacement of 0.5 mm for PC-ABS composites with 0.75 and 1.0 wt.% CNTs.

**Table 1 micromachines-10-00063-t001:** Surface resistance per length unit measured for the laser tracks obtained under optimized conditions (P = power; S = scan rate; N = number of repetitions; F = frequency; D = defocus) on several polymer–carbon filler systems (* not good reproducibility).

Material	Filler	Production Process	Laser Parameters	Surface Resistance per Length Unit
HDPE/6 wt.% MWCNTs	MWCNTs Nanocyl NC7000	Masterbatch produced by melt compounding high-density polyethylene (HDPE) and multiwall carbon nanotubes (MWCNTs)	P = 10%, S = 100 mm/s, N = 25, F = 15 kHz, D = 50 mm	1.28 kΩ/cm
HDPE/4 wt.% MWCNTs	MWCNTs Nanocyl NC7000	Twin screw extrusion of masterbatch HDPE/MWCNTs and HDPE, pelletizing and injection molding	P = 10%, S = 100 mm/s, N = 25, F = 15 kHz, D = 50 mm	19.7 kΩ/cm
HDPE/4 wt.% MWCNTs/4 wt.%GNPs	MWCNTs Nanocyl NC7000; GNPs ABCR 25 µm 6–8 nm	Twin screw extrusion of masterbatch HDPE/MWCNTS and masterbatch HDPE/graphene-like nanoplatelets (GNPs), pelletizing and injection molding	P = 10%, S = 100 mm/s, N = 25, F = 15 kHz, D = 50 mm	46 kΩ/cm
HDPE/4 wt.% MWCNTs/4 wt.% graphite	MWCNTs Nanocyl NC7000; Graphite Alfa-Aesar 7–10 µm	Twin screw extrusion of masterbatch HDPE/MWCNTs and masterbatch HDPE/graphite, pelletizing and injection molding	P = 10%, S = 100 mm/s, N = 25, F = 15 kHz, D = 50 mm	7.01 kΩ/cm
PP/30 wt.% biochar	Biochar pellets OSR700 UK Biochar Research Center	Melt blending of PP and biochar, twin screw extrusion, pelletizing and injection molding	P = 15%, S = 50 mm/s, N = 7, F = 5 kHz, D = 30 mm	4 MΩ/cm (antistatic)
PP/2 wt.% CNTs	MWCNTs Nanocyl NC7000	Melt blending of masterbatch PP-MWCNTs and PP, pelletizing and injection molding	P = 20%, S = 50 mm/s, N = 25, F = 10 kHz, D = 200 mm	0.9 kΩ/cm
PP/1 wt.% CNTs	MWCNTs Nanocyl NC7000	Melt blending of masterbatch PP-MWCNTs and PP, pelletizing and injection molding	P = 20%, S = 200 mm/s, N = 25, F = 15 kHz, D = 100 mm	12.3 kΩ/cm
PC-ABS/1.0 wt.% CNTs	MWCNTs Nanocyl NC7000	Twin screw extrusion of masterbatch PC-ABS-MWCNTs and PC-ABS, pelletizing and injection molding	P = 5%, S = 300 mm/s, N = 30, F = 30 kHz, D = 0 mm	3.96 kΩ/cm
PC-ABS/0.75 wt.% CNTs	MWCNTs Nanocyl NC7000	Twin screw extrusion of masterbatch PC-ABS-MWCNTs and PC-ABS, pelletizing and injection molding	P = 5%, S = 100 mm/s, N = 20, F = 5 kHz, D = 0 mm	0.41 kΩ/cm
PC-ABS/0.5 wt.% CNTs	MWCNTs Nanocyl NC7000	Twin screw extrusion of masterbatch PC-ABS-MWCNTs and PC-ABS, pelletizing and injection molding	P = 10%, S = 100 mm/s, N = 20, F = 30 kHz, D = 0 mm	0.02 kΩ/cm
PP/5 wt.% GNPs	GNPs ABCR (1–2 µm)	Melt mixing, pelletizing and injection molding	P = 20%, S = 200 mm/s, N = 25, F = 15 kHz, D = 100 mm	≈5 * kΩ/cm
ABS/5 wt.% GNPs	GNPs ABCR (1–2 µm)	Melt mixing, pelletizing and injection molding	P = 20%, S = 200 mm/s, N = 25, F = 15 kHz, D = 100 mm	≈5 * kΩ/cm

**Table 2 micromachines-10-00063-t002:** Resistance variation occurring during a single cycle of deformation and the average resistance value of a portion of the bar 40 mm long (displacement of up to 1.5 mm, displacement speed: 50 mm/min) after increasing the number of cycles.

Material (CNTs wt.%)	Resistance Variation (%)			Average Resistance (kΩ)		
	Cycle 1	After 300 Cycles	After 1000 Cycles	Cycle 1	After 300 Cycles	After 1000 Cycles
3	0.22	0.50	0.50	31.652	31.690	31.690
4	0.15	0.80	0.80	0.732	0.733	0.733
5	0.25	0.90	0.90	0.188	0.189	0.190

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
