# Peer review of "Laser Treatments for Improving Electrical Conductivity and Piezoresistive Behavior of Polymer–Carbon Nanofiller Composites"

_micromachines, 2019, doi:10.3390/mi10010063_

Reviewer 1 Report

In this work, the authors present the results for the effect of laser treatment on the surface conductivity of composite materials. I have a few questions that must be answered before acceptance.

1) It is stated that the laser treatment causes the pyrolysis of the matrix without modifying the reinforcements, this statement must be supported by data or previous works.

2) It is clear nowadays that when adding carbonaceous fillers to a polymeric matrix a network is formed, where the vertices are the reinforcements. Hence the conduction is due to hopping or tunneling between the fillers when it is above the percolation threshold. How the evaporation of the matrix would change this conductive network?

3) Using the data figure 6 the critical exponent (t) should be calculated and compared with other works.

Author Response

We have carefully revised our paper according to all the Reviewers’ comments.  For sake of clarity we have reported in the following all these comments and, after each of them, the explanation of the revisions we have done in order to give a response to them. In the following responses we have indicated the lines of the revised manuscript where each revision appears. All the revisions are put in evidence in the emended manuscript by yellow colour to make easy to find them. 

Reviewer 1

Point 1

It is stated that the laser treatment causes the pyrolysis of the matrix without modifying the reinforcement, this statement must be supported by data or previous works.

Answer

It is well known that laser treatments can be used for improving the quality of CNTs (single wall nanotubes in particular). These treatments can result in impurity removal and defect healing/recrystallization and then in changing of optical properties and electrical resistance. On the other hand, these treatments have been adopted for improving the characteristics of CNTs films deposited on substrates (also polymeric) and the result of the treatment sometimes was correlated with the interaction between substrate and film. In addition, the treatment conditions adopted to this purpose seem very different from those here used for processing the conductive tracks (wavelength used for the above mentioned treatments one order of magnitude lower than that adopted for processing the conductive tracks). Anyway, we agree it would be useful to mention the possibility that the laser could also affect the characteristic of the fillers. This subject has been discussed at lines 158-167 of the revised paper.

Point 2

It is clear nowadays that when adding carbonaceous fillers to a polymeric matrix a network is formed, where the vertices are the reinforcements.  Hence the conduction is due to the hopping or tunneling between the fillers when it is above the percolation threshold. How the evaporation of the matrix would change this conductive network?

Answer

This remark shows as we were not formerly able to explain well this subject in our paper. The tunneling effect allows to obtain only little conductivity when the filler concentration is still below the percolation threshold and then when each filler particle is not too far from its neighbors.  Only when the conductive particles get in touch the percolation threshold is achieved and fine conductivity can be observed.

Most of the composites submitted to the laser writing process were insulating materials, with a conductive filler concentration below the percolation threshold.  In those causes conductive network formed locally only where the laser beam caused the matrix evaporation.

This mechanism has been better described in the revised version (see lines 177-184)

Point 3

Using the data Figure6 the critical exponent (t) should be calculated and compared with other works.

Answer

We were not able to find in the literature data about the percolation curve for our composite made of a complex matrix consisting in a blend co-polymer EPDM and PP (with 60:40 weight ratio) filled by CNTs, but a comparison of its percolation threshold can be done with literature data concerning composites PE/CNTs and PP/CNTs and PP-EPDM/CNTs.

This comparison has been done in the revised version at lines 311-315.

Reviewer 2 Report

In this paper, the electrical properties of polymer composites containing carbon nanofillers were investigated after repeated CO2 laser irradiation. The content is interesting, but it seems that the paper is not yet ready for publication in this journal. I have listed some concerns below.

1. It is too difficult to read the introduction. Please split the paragraph in the introduction part.

2. The authors missed some important recent papers (for example, T. Kim et al., ACS Nano, 10(4), 4770-4778, 2016). Also, most of references cited in this paper are not representative.

3. Please use a micrometer unit to describe the wavelength of the CO2 laser.

4. Please let us know the beam spot size and the energy of the single pulse of your laser.

5. Figure 1 does not intuitively present the concept of this work. I do not know what green and sky blue bars mean. There is no description of the SEM image and there is no scale bar.

6. What does the x-axis in Figure 2 represent? How was the pareto plot deduced? Figure 2 does not seem to be needed.

7. Error bars are required in Figure 3.

8. Figures 4 and 5 can be merged.

9. I cannot understand why the authors tested various polymer matrices (HDPE, PP, PC-ABS, ABS, EPDM, etc). Variables are too messy and confusing. If the authors want to show the tendency according to the laser irradiation parameters, it is better to select only one polymer matrix and track the electrical and morphological changes before and after the laser irradiation.

10. Overall, reconstruction of the figures and table is required.

11. The author should provide the absolute value of the percolation threshold and compare the values before and after the laser irradiation for all cases.

12. The authors should test the piezoresistive properties with more severe deformation.

Author Response

We have carefully revised our paper according to all the Reviewers’ comments.  For sake of clarity we have reported in the following all these comments and, after each of them, the explanation of the revisions we have done in order to give a response to them. In the following responses we have indicated the lines of the revised manuscript where each revision appears. All the revisions are put in evidence in the emended manuscript by yellow colour to make easy to find them. 

Reviewer 2

Point 1

 It is too difficult to read the introduction. Please split the paragraph in the introduction part.

Answer

We have revised the introduction by splitting it in paragraph (see lines 28, 60, 84, 97) and trying to make more clear the aim of this work as an evolution of the state of the art (see lines 99-101).

Point 2

The authors missed some important recent papers (for example, T. Kim et al. ACS Nano, 10 (4), 4770-4778, 2016).  Also, most of the reference cited in this paper are not representative.

Answer

Owing to the enormous number of papers on development of polymer-based composites with carbon nanofiller published during the last years and in order to limit the number of references, we chose to cite mainly recent reviews that summarize the most relevant advances in the field of functional nanocomposites. Hopefully the meaning of these references should be more clear after the re-organization of the introduction in sub-sections (e.g. see lines 32-35). Anyway, according to the suggestion of both reviewers we have added more references (see in the revised paper in particular Ref. 4).

Point 3

Please use a micrometer unit to describe the wavelength of the CO2 laser.

Answer

Done (line 131)

Point 4

Please let us know the beam spot size and the energy of the single pulse of your laser.

Answer

The spot size with defocusing 0 has been reported  (see line 131).

The energy furnished by any single pulse is the percentage of the maximum power (100 W) indicated for the various laser treatments (line 272).

Point 5.

Figure1 does not intuitively present the concept of this work.  I do not know what green and sky blue bars mean. There is no description of the SEM image and there is no scale bar.

Answer

A comment to Figure1 has been added (see lines 177-184) to clarify the meaning of this Figure.  The scale bars have been added to the SEM micrographs shown in this Figure, and the Figure caption has been revised (lines 187-188).

Point 6.

What does the x-axis in Figure2 represents? How was the Pareto plot deduced? Figure2 does not seem to be needed.

Answer

Pareto plot is one of the outcomes obtained when the DOE approach is adopted. A reference concerning this method has been given in the revised paper (Ref. 24), more details have been given about the experimental method (lines 193-195) and the meaning of Pareto plot has been explained (see lines 198-203).  Figure2 shows as there is an enormous difference between the effect of the laser writing parameters on the conductivity of the tracks (lines 207-210). However also the parameters that have negligible effect on resistance (as showed by Pareto plot) must be considered because they affect the track morphology (see new Figure 4)

Point 7.

Error bars are required in Figure3.

Answer

The curves in this Figure were obtained by repeating on the same track the measurement of resistance after increasing numbers of laser runs. In practice the laser treatment was interrupted for the measurement and then re-started several times.  In this manner it was possible to appreciate the change of resistance only depending on the number of repetitions. Each point of the curves is not the result of several measurements performed on different samples but it comes from a measurement performed on the same track after different repetitions. For this reason the data cannot be treated in a statistical way for calculating the error bars.  The experimental method adopted for obtaining the curves in Figure3 is better explained in the text (see lines 214-215).

Point 8.

Figures 4 and 5 can be merged.

Answer

A merge of these Figure has been presented in the revised paper.

Point 9.

I cannot understand why the authors tested various matrices (HDPE, PP, PC-ABS, ABS, EPDM, etc). Variables are too messy and confusing. If the authors wants to show the tendency according to the laser irradiation parameters, it is better to select only one polymer matrix and track the electrical and morphological changes before and after the laser irradiation.

Answer

Several composites differing for both matrix and kind of filler were testes with the aim of showing that the laser writing process can be successfully applied on most of these composites. This was stated in the introduction (lines 99-101), the discussion (lines 258-261 and 285-287) and the conclusions (lines 387-390).   This was the main result we would like to underline. However, also the evaluation of the different relevance on the conductivity of the various processing parameters has been presented by using Pareto Plot (see comments to point 6).  Nevertheless, it is worth of notice that also the parameters that exert a minor effect on the electrical resistance have some effect on the morphology of the conductive tracks (see new Figure4). Since for practical applications there are constraints on the morphology of the tracks (because of the possible formation of short circuits) also these last parameters should be optimized (see lines 255-262). For the reasons mentioned above there is not a set of parameters of general validity and then a general tendency that can be adopted for every kind of composite. This outcome is put in evidence in table 1 and discussed in the revised version at lines 272-277.

Table 1 also shows that after having tailored the processing parameters to each specific system the conductivity of the different conductive tracks is not the same.  This means that different systems provide conductive tracks with different electrical resistance, in spite of the optimization of the laser treatment. 

To select only one matrix/filler system, as the reviewer suggests, would not permit to get to the conclusions exposed above and describe properly the subject under investigation. 

We have revised the relevant part of the manuscript with the aim to make more clear to the readers this matter.

Point 10.

Overall reconstruction of Figures and table are required

Answer

Several revisions have been done on Figures and, above all, their meaning was better explained in the text.

Point 11.

The authors should provide the absolute value of the percolation threshold and compare the value before and after the laser irradiation for all cases.

Answer

Careful examination of literature about the threshold for electrical conductivity of many kind of polymer matrix-nanofiller composites clearly brings to the conclusion that the threshold is heavily affected by the quality of nanofiller (purity of CNTs, size of GNPs, etc) as well as by the composite production method (see introduction, lines 48-58 and 61-64). As a consequence, only indications about the range of filler concentration suitable for achieving the percolation threshold (but not an absolute value) can be given for each kind of composite on the base of the literature. On the basis of resistance measurements we assessed that most of the composite (but not all of them) we produced for laser treatment showed a filler concentration below the threshold (lines 263-271). Anyway, when useful for the result discussion, literature data concerning the filler concentration at the percolation threshold reported in the literature have been mentioned (lines: 268-271; 311-315).

The threshold for conductivity can be defined as the minimum filler concentration that allows for the formation of a conductive network and a sudden jump of conductivity, and applies to composites with an homogeneous distribution of filler inside the matrix. In other words for each composite system and for each processing method the threshold is related to the variation of a single parameter (the filler concentration).

Laser irradiation alters locally the distribution and concentration of filler starting from an homogeneous composite.  For this reason the concept of percolation threshold is not meaningful after laser irradiation.

In fact the laser irradiation can be unsuccessful for achieving locally conductivity (when the treatment parameters are not appropriate), and there are a lot of possible parameter combinations that can give this bad result.

When the laser irradiation treatment is successful, instead, the laser track shows concentration of filler always above the percolation threshold. However there are many possible combinations of laser parameters that bring to the achievement of conductivity, and therefore there is not a single condition for exceeding the percolation threshold but a lot.

Point 12.

The authors should test the piezoresistive properties with more severe deformation.

Answer

In our opinion piezoresistivity of polymer-based composites offers the chance of integrating switches into  semistructural components, and improve their piezoresistive performance with the help of laser functionalization.  For example this could find application in a control panel of an electrical appliance or a dashboard of a car (please note that this work was done in cooperation with a car producer).  For these applications the switch should be sensitive enough to give a response to a pressure exerted by a finger touch. The action of a finger on a semistructural component is expected to give rise to limited deformations. In addition only elastic deformations, and forces  not exceeding the yield limit, should be considered as a component must not be deformed in a permanent manner (see lines 321-325). For these reasons a displacement ranging between 0.5 and 2.5 mm seemed appropriate to us.

Anyway, we agree that for other possible applications it could be interesting to investigate the piezoresistivity under more important deformations.

Round  2

Reviewer 1 Report

I recommend the publication of the article in the present form.

Reviewer 2 Report

I would like to thank the authors for making the revision. The manuscript has obviously been improved and is now acceptable.